# Degradation of Dioxins and DBF in Urban Soil Microcosms from Lausanne (Switzerland): Functional Performance of Indigenous Bacterial Strains

**DOI:** 10.3390/microorganisms13102306

**Published:** 2025-10-05

**Authors:** Rita Di Martino, Mylène Soudani, Patrik Castiglioni, Camille Rime, Yannick Gillioz, Loïc Sartori, Tatiana Proust, Flavio Neves Dos Santos, Fiorella Lucarini, Davide Staedler

**Affiliations:** 1TIBIO Suisse Romande, Chemin de Bérée 4C, 1010 Lausanne, Switzerland; 2Group for Functionalized Biomaterials, Institute of Chemical Sciences and Engineering, Ecole Polytechnique Fédérale de Lausanne, 1015 Lausanne, Switzerland; 3TIBIOLab Sàrl, Route d’Yverdon 34, 1373 Chavornay, Switzerland; 4Department of Biomedical Sciences, University of Lausanne, Rue du Bugnon 27, 1011 Lausanne, Switzerland; 5School of Engineering and Architecture of Fribourg, Institute of Chemical Technology, HES-SO University of Applied Sciences and Arts of Western Switzerland, Boulevard de Pérolles 80, 1700 Fribourg, Switzerland

**Keywords:** dioxins, bioremediation, soil pollution, indigenous bacteria, dibenzofuran, co-culture, urban soil, 2,7-dichlorodibenzo-*p*-dioxin

## Abstract

Urban soils are often affected by long-term deposition of persistent organic pollutants, including polychlorinated dibenzo-*p*-dioxins (PCDDs) and dibenzofurans (PCDFs). This study evaluated the biodegradation potential of indigenous bacterial strains isolated from chronically contaminated soils in Lausanne, Switzerland. Using selective enrichment techniques, five strains were isolated, with no biosafety concerns for human health and environmental applications. These isolates were screened for their ability to degrade dibenzofuran (DBF) and 2,7-dichlorodibenzo-*p*-dioxin (2,7-DD) under mineral medium conditions. A simplified two-strain consortium (*Acinetobacter bohemicus* and *Bacillus velezensis*) and a broader five-strain co-culture were then applied to real soil microcosms over a 24-week period. This work provides the first experimental evidence that *A. bohemicus* and *B. velezensis* can degrade DBF and 2,7-DD under controlled conditions. Dioxin concentrations were monitored at 4, 8, and 24 weeks using a Gas Chromatography Mass Spectrometry (GC-MS). In laboratory conditions, co-cultures showed enhanced degradation compared to individual strains, likely due to metabolic complementarity. In soil, the simplified two-strain consortium performed better at dioxin degradation, especially at earlier time points. Although no statistically significant reductions were observed due to high variability and limited sample size, consistent trends emerged, particularly at the most contaminated site. These findings support the relevance of testing bioremediation strategies under realistic environmental conditions.

## 1. Introduction

Dioxins are highly toxic, persistent organic pollutants (POPs) that pose significant environmental and human health risks due to their extraordinary chemical stability, high bioaccumulation potential, and biomagnification within food webs [1,2]. Chemically, these compounds include polychlorinated dibenzo-*p*-dioxins (PCDDs), polychlorinated dibenzofurans (PCDFs), and certain polychlorinated biphenyls (PCBs) [2,3]. They are mainly formed through incomplete combustion processes, industrial manufacturing, chlorine-based chemical production, and waste incineration [1,2,4]. Dioxins are associated with multiple adverse health outcomes, including carcinogenesis, immunotoxicity, reproductive and developmental disorders, endocrine disruption, and neurological impacts [2,3,5,6]. Their persistence in environmental matrices such as soils and sediments can span several decades, presenting substantial long-term contamination challenges globally [2,7,8].

Historical events, such as the Seveso disaster and the extensive use of dioxin-contaminated herbicides during the Vietnam War, have significantly heightened global awareness regarding the implications of dioxin pollution [9,10,11]. Significant contamination persists in many regions despite the establishment of international regulatory frameworks aimed at limiting dioxin emissions, like the Stockholm Convention [11,12]. A notable case is found in Lausanne, Switzerland, where extensive dioxin contamination was detected around a former waste incineration plant active between 1958 and 2005 [13]. Soil samples from this site showed high dioxin concentrations, with levels reaching up to 450 ng TEQ (Toxic Equivalents)/kg, posing potential ecological and long-term health risks to local populations [13,14]. Concentrations were measured for individual dioxin and furan congeners and aggregated into TEQ values using WHO 2005 toxic equivalency factors (TEFs). TEQ represents the sum of congener concentrations weighted by their respective TEFs, providing a standardized estimate of the total dioxin-related toxicity in each sample [15,16,17]. Although initial public health assessments did not indicate a significant impact on the local population, the persistent soil contamination represents a serious ecological and economic burden that demands innovative remediation solutions [13,14].

Bioremediation is an approach that utilizes microorganisms or plants to degrade environmental pollutants, and it is a potentially sustainable and cost-effective solution to address dioxin contamination [18,19,20,21,22,23]. Various microbial species have demonstrated substantial dioxin-degrading capacities [18,21]. Aerobic bacteria such as *Sphingomonas wittichii* and anaerobic organohalide-respiring bacteria like *Dehalococcoides* spp. degrade dioxin through dioxygenase-mediated oxidative cleavage and reductive dechlorination, respectively [24,25,26]. However, while these biochemical pathways show promise, the implementation of bioremediation technologies at field-scale remains challenging due to slow degradation kinetics, incomplete pollutant breakdown, the accumulation of toxic intermediate products, and difficulty scaling laboratory findings to complex field conditions [21,27].

Recent research has highlighted the potential advantages of employing indigenous, site-specific microbial communities for bioremediation [22,28]. Native microbial populations that have been chronically exposed to contaminated environments typically demonstrate superior survival rates and pollutant-degradation efficiency compared to externally introduced strains [21,29]. Biostimulation strategies can significantly enhance the inherent degradation pathways of these endogenous microbial communities: the optimization of nutrient concentrations, moisture levels, oxygen availability, and other key physicochemical soil parameters can lead to more efficient and environmentally compatible remediation outcomes [23,30].

In line with these findings, the current study employed a laboratory-based enrichment strategy using original contaminated soils from Lausanne to select native bacterial strains capable of degrading dioxins. Two model compounds were chosen based on literature evidence: dibenzofuran (CAS 132-64-9; DBF), known for its efficacy in selecting microbial strains capable of biodegrading polychlorinated dibenzofurans [22,31,32,33,34], and 2,7-dichlorodibenzo-*p*-dioxin (CAS 33857-26-0; 2,7-DD), representing polychlorinated dibenzo-*p*-dioxins (Figure 1) [35].

DBF was selected because of its structural similarity to dioxins and its capacity to serve as a carbon source without presenting significant toxicity issues during laboratory handling [22,36]. Conversely, 2,7-DD contains chlorine atoms, thus it was used to select strains capable of using chlorinated dioxins both as a carbon source and through redox reactions involving C-Cl bonds [22,37]. Microbe enrichment and selection were first carried out under controlled laboratory conditions. Subsequently, the degradation potential of the isolated strains was tested on real, chronically contaminated soils from Lausanne, providing insight into selected strains functionality in more complex environments.

This work culminated in soil microcosm experiments conducted under semi-natural conditions, where the biodegradation capacity of the selected strains was monitored over an extended period of up to six months. These trials aimed to evaluate the persistence and efficacy of bacterial treatments in realistic matrices and under operationally relevant time frames [18,38,39,40,41].

This study was designed with potential real-world applications in mind, and thus it specifically addresses critical biosafety parameters. It involves the use of bacterial strains strictly belonging to Risk Group 1 according to EU and US classification, defined as organisms posing negligible or no risk to human health or the environment, and being non-pathogenic and non-genetically modified [42,43]. The utilization of Risk Group 1 microorganisms is a fundamental principle in in situ bioremediation, ensuring the environmental safety and public health compliance essential [44,45]. Moreover, this study includes the verification of antibiotic resistance profiles, as ensuring the absence of significant antibiotic resistance is essential to avoid unintended ecological impacts and to preserve environmental microbiota integrity [46]. Additionally, this research incorporates scaling up the production of isolated bacteria through verification of phenotype stability, demonstrating that degradation capabilities can be maintained even after biomass amplification in non-selective conditions—a critical step for field-scale implementation [47,48].

## 2. Materials and Methods

### 2.1. Reagents

Dibenzofuran (CAS 132-64-9; DBF) and 2,7-dichlorodibenzo-*p*-dioxin (CAS 33857-26-0; 2,7-DD) were purchased from Sigma-Aldrich, Burlington, MA, USA. Tryptic Soy Broth (TSB) and Tryptic Soy agar (TSA) used for bacteria cultures were purchased from BIOKAR diagnostics. Recipes for other bacterial media are listed in sections further below, along with their component suppliers.

### 2.2. Soil Samples

Soil samples were collected from three distinct sites in Lausanne, Switzerland—Signal (SIG), Ancien Stand (AST), and Eracom (ERA)—each exhibiting a characteristic profile of dioxin contamination (Appendix A). All sites contained measurable levels of polychlorinated dibenzo-*p*-dioxins (PCDDs) and polychlorinated dibenzofurans (PCDFs), with substantial variation in congener concentrations. The SIG site showed the highest contamination, with several congeners—such as octachlorodibenzo-*p*-dioxin (OCDD), 1,2,3,4,6,7,8-heptachlorodibenzo-*p*-dioxin (HpCDD), and HpCDF—exceeding 400 ng/kg. AST presented intermediate contamination levels among the three sites, while ERA exhibited the lowest, with most congeners below 100 ng/kg. This gradient of contamination was used to evaluate the performance of bioremediation treatments under increasingly challenging environmental conditions.

Soil sampling was carried out in two campaigns, one in May 2023 and another in May 2024. For both campaigns, samples were taken from the top 30 cm of soil and stored at ambient temperature in sealed 10 kg buckets. During the first campaign, approximately 20 kg of soil per site were collected, and part of it was used for the preparation of the initial inocula for microbe selection (see Section 2.3 Soil Inoculum Preparation). Additional 90 kg per site were collected over the second campaign to assess dioxin bioremediation potential in a setting closer to field conditions (see Section 2.11 Treatment of real soil samples for bioremediation experiments). Detailed addresses, geographic coordinates, and full concentration data (ng/kg) are provided in Appendix A.

### 2.3. Soil Starting Inoculum Preparation

Soil samples collected during the first sampling campaign from the three contaminated sites (SIG, AST, ERA) were processed independently to preserve their distinct indigenous microbial communities. Each starting inoculum was prepared by suspending 3 g of contaminated soil from each contaminated site separately in 200 mL of sterile solutions specifically formulated to support microbial survival. For bacterial isolation, a mineral medium containing 6.4 g/L Na_2_HPO_4_·7H_2_O, 1.5 g/L KH_2_PO_4_, 0.3 g/L NaCl, 0.5 g/L NH_4_Cl, 0.5 g/L (NH_4_)_2_SO_4_, 0.5 g/L MgSO_4_·7H_2_O, 0.1 g/L CaCl_2_, and 0.001 g/L FeSO_4_·7H_2_O was used (all from Sigma-Aldrich). For fungal isolation, a different mineral medium was employed, consisting of 2 g/L K_2_HPO_4_, 0.5 g/L MgSO_4_·7H_2_O, 0.5 g/L KCl, 0.018 g/L FeSO_4_·7H_2_O, and 2 g/L NaNO_3_ (all from Sigma-Aldrich). The resulting suspensions were incubated for a week at 20 °C, 130 rpm to allow microbes transfer from the soil to the surrounding liquid medium. These samples served as starting inocula for subsequent enrichment cultures aimed at selecting microbial strains capable of biodegrading dioxins.

### 2.4. Selection of Dioxin-Degrading Microbial Strains

The selection of microbial strains capable of degrading dioxins was carried out using a successive enrichment approach. The enrichment process employed DBF as the primary selective substrate, chosen for its structural similarity to dioxins and its relatively low toxicity [20,28,31,32,34,35]. DBF was added directly to the bacterial mineral media at concentrations of 500 mg/L for the enrichment phases, progressively becoming the sole available carbon source as the selective pressure increased. This condition was optimized based on existing literature to minimize acute toxicity while promoting bacterial growth and selection [21,49]. Enrichments were performed in six successive cycles of 3 weeks each. For the first cycle, 1 mL of starting inoculum was transferred into three flasks with 200 mL of fresh mineral medium supplemented with DBF 500 mg/L. This process was performed separately for the three contaminated sites (SIG, AST, ERA) to continue the preservation of the distinct indigenous microbial communities. At each following cycle, 1 mL of the previous culture was transferred into 200 mL of fresh mineral medium containing DBF at the same concentration. Bacterial viability was monitored at every cycle through microscope cell counting to ensure active culture propagation (see Section 2.6 Quantification and Identification of Bacterial Strains). DBF microbial degradation capacity was assessed at the end of the final DBF enrichment cycle (cycle six) by quantifying pollutant reduction and measuring microbial biomass growth. Subsequently, a single additional enrichment step was carried out transferring 1 mL of cycle six culture into 200 mL of mineral medium supplemented with 5 mg/L of the chlorinated dioxin compound 2,7-DD. After 3 weeks, 2,7-DD microbial degradation capacity was assessed too. At the end of each enrichment cycle, 10 mL of culture were collected and stored at 4 °C for both microbial analysis and back-up sample for the preparation of further enrichments whether needed.

### 2.5. Chemical Analysis of DBF and 2,7-DD

To quantify the microbial treatment effects, analytical methods were developed for the quantification of the reference compounds DBF and 2,7-DD. From the initial culture volume of 200 mL, 10 mL were collected for microbiological analyses, while the remaining 190 mL were extracted three times with 10 mL of ethyl acetate each time. The combined extracts were then brought to a final volume of 40 mL and filtered through glass wool. Subsequently, 30 mL of the filtered extract were evaporated and reconstituted with 1 mL of ethyl acetate for injection into the analytical instrument.

The efficiency of the extraction method was assessed by adding known quantities of analytical standards in the sample prior extraction. Extracted calibration curves (0.5–100 mg/L) were systematically prepared for quantification, and acceptable recovery rates were maintained within the range of 70% to 130% with limit of quantification (LOQ) of 0.5 ppm. Gas chromatography-mass spectrometry (GC-MS) analysis was performed using a GCMS-QP2010 Ultra (Shimadzu Corporation, Kyoto, Japan) equipped with an OPTIMA-5 MS column (30 m length, 0.25 mm diameter, 0.25 µm thickness) and operated using LabSolutions GC-MS Shimadzu software. Initial method development involved full-scan analyses to identify optimal parameters for single ion monitoring (SIM) mode quantification. For SIM analysis, the oven temperature was programmed as follows: an initial temperature of 80 °C was increased at a rate of 20 °C/min to 200 °C, followed by an increase of 5 °C/min to 290 °C, and a final increase of 20 °C/min to 300 °C, which was maintained for 5 min. The injection temperature was set at 250 °C and injections were performed in splitless mode. Helium was employed as the carrier gas with a total flow rate of 20 mL/min. Specific mass transitions, retention times, limit of detection and limit of quantification for DBF and 2,7-DD quantification are provided in Appendix A.

### 2.6. Quantification, Isolation and Identification of Degrading Microbial Strains

Microbes were initially quantified using a counting chamber (BLAUBRAND^®^ Neubauer improved) under a microscope to determine bacterial cell density (Leica Microscope DM1000 and Leica Microscope Objective N Plan 40x/0.65 PH2 506099, Leica, Wetzlar, Germany). Additionally, bacterial growth was monitored by measuring optical density at 600 nm (OD600) using a spectrophotometer (DeNovix DS-C Spectrophotometer, DeNovix, Wilmington, DE, USA). Individual colonies were isolated after the last enrichment step by plating 0.1 mL of enrichment cultures onto TSA to select bacteria. Plates were incubated at 25 °C and colonies formation was monitored every day for 1 week.

Identification of the isolated colonies was subsequently carried out using Matrix-Assisted Laser Desorption/Ionization—Time of Flight (MALDI-TOF) mass spectrometry, performed externally by Mabritec AG (Riehen, Switzerland). The resulting identified species were then compared against Swiss, European, and American microbial risk classifications (Risk Group 1 to 4). Only microbial strains classified as Risk Group 1—non-pathogenic, non-genetically modified organisms presenting negligible or no risk to human health and the environment—were selected for further verification and potential bioremediation applications [44,45].

### 2.7. Degradation Assays with Individual Strains and Defined Co-Cultures

Biodegradation tests were conducted to assess DBF and degradation capacity of the selected strains in both mono- and co-culture. All assays were performed in sterile mineral medium supplemented with DBF and 2,7-DD at concentration of 5 mg/L each. Cultures were incubated for 3 weeks at 20 °C under agitation (130 rpm). The total assay volume was 20 mL. For monoculture experiments, each strain was inoculated with 200 μL of cell suspension at 10^9^ CFU/mL, while for co-culture experiments 40 μL of each strain suspension at 10^9^ CFU/mL were added (total 200 μL), maintaining the same overall inoculum density of 10^7^ cells/mL. DBF and 2,7-DD degradation were quantified by GC-MS at the end of the incubation period.

### 2.8. Antibiotic Susceptibility Testing

Antibiotic resistance profiles of the selected bacterial strains were quantitatively assessed by determining the minimum inhibitory concentration (MIC) using the ETEST^®^ method (bioMérieux SA, Lyon, France). The susceptibility tests covered 14 antibiotics from various classes, including beta-lactams, penicillins, cephalosporins, fluoroquinolones, macrolides, aminoglycosides, carbapenems, tetracyclines, glycopeptides, and oxazolidinones, encompassing both bactericidal and bacteriostatic agents (Appendix A). ETEST^®^ strips provided a gradient of 15 antibiotic concentrations ranging from 0.002 to 256 µg/mL, depending on the specific antibiotic. Overnight cultures of the selected strains were prepared in TSB (25 °C, 130 rpm), and the optical density at OD600 was measured. Cultures were standardized to an OD600 of 0.5 using phosphate-buffered saline (PBS: 8 g/L NaCl, 2.72 g/L Na_2_HPO_4_ · 7 H_2_O, 0.245 g/L KH_2_PO_4,_ 0.2 g/L KCl, all from Sigma-Aldrich), and aliquots of 150 μL were spread on TSA plates to have a uniform bacterial lawn. One ETEST^®^ strip was carefully placed onto one lawn plate, and the procedure was individually repeated per every antibiotic and per every strain. Bacterial sensitivity was determined by observing inhibition zones around the strips after 16–24 h incubation at 25 °C. MIC values, representing the lowest antibiotic concentration inhibiting bacterial growth, were recorded at the intersection point of bacterial growth inhibition with the strip gradient.

### 2.9. Scale-Up in 2 L Cultures: Contamination Assessment

Bacterial scale-up was performed using TSB as a general cultivation medium. Five 2 L flasks were inoculated with one strain each and they were incubated at 25 °C, 130 rpm for 50 h. During incubation time, bacterial growth and metabolic activity were manually monitored at regular intervals by measuring OD600, pH, conductivity, number of cells/mL. Values of pH were determined using a Mettler Toledo SevenCompact, Mettler-Toledo International Inc., Greifensee, Switzerland, and conductivity was measured using a CO 3100H—Conductivity Meter (VWR), VWR International BV, Leuven, Belgium. At the end of the incubation time, standard agri-food domain microbiological screenings were performed to detect potential contamination by bacterial pathogens harmful to humans that could have occurred in this large batch culture conditions. These analyses were conducted by Eurofins Scientific (Lausanne, Switzerland). The tested parameters included the presence of: *Bacillus cereus*, coagulase-positive *Staphylococcus* spp., *Campylobacter* spp., *Listeria monocytogenes*, *Salmonella* spp., *Escherichia coli* and coliform bacteria, fecal enterococci, and *Pseudomonas aeruginosa* (Appendix A).

### 2.10. Phenotype Stability Assay Following Scale-Up

To evaluate whether the pollutant-degradation phenotype was retained after growth in non-selective conditions, all strains were re-inoculated into selective medium following the 2 L TSB scale-up. For each strain, 1 mL of culture from the 2 L flask was transferred into 200 mL of sterile mineral medium supplemented with 10 mg/L of 2,7-DD and 100 mg/L of DBF. Each strain was tested individually, and a defined co-culture containing all five strains in equal proportions (20% each) was also included in the experiment. All cultures were incubated at 25 °C, 130 rpm for 4 weeks. DBF and 2,7-DD degradation were measured at the end of the incubation period using GC-MS as described previously. These results were compared with degradation results of the same strains that were exclusively cultured in mineral medium containing DBF and 2,7-DD as sole carbon sources. This test was designed to assess the ability of the strains to maintain their functional phenotype after biomass production in rich, non-selective medium.

### 2.11. Treatment of Real Soil Samples for Bioremediation Experiments

Bioremediation treatments were conducted using non-sterilized soil samples collected over the second sampling campaign from three contaminated sites (SIG, AST, ERA). For each experimental condition, three independent replicates were employed, each consisting of 10 kg of soil per site. Additionally, one untreated 10 kg sample per site served as negative control. Prior to treatment application, each 10 kg soil sample was thoroughly homogenized using an electric mixer suitable for this scale. Two experimental treatments were conducted: the first involved inoculation with 250 mL of cell suspension at 10^8^ CFU/mL of the two bacterial strains exhibiting the highest degradation performance, while the second treatment consisted of 100 mL of cell suspension at 10^8^ CFU/mL of a defined five-strain co-culture. Both treatments were applied at a final concentration of 10^7^ CFU/g soil. Control samples received an equivalent volume of H_2_O to match moisture conditions provided by bacterial inoculations. The same electric mixer was used to apply and uniformly distribute treatments throughout the soil samples. All treated samples were incubated at ambient temperature in ventilated, static systems maintained at ambient humidity, closely mimicking realistic field conditions. To ensure consistent microbial distribution and homogeneity, soils were thoroughly mixed prior to each sampling time point. Samples were collected at 4, 8 and 24 weeks post-treatment to monitor the evolution of pollutant degradation processes over time. Dioxin concentrations in soil samples were determined externally by SolConseil (Gland, Switzerland), an ISO 17025-certified analytical laboratory, using a standard Soxhlet-based method according to the method DIN EN 16190 (2019-10). The analysis included a comprehensive panel of dioxin congeners, covering tetra- to octa-chlorinated dibenzo-*p*-dioxins (PCDDs) and dibenzofurans (PCDFs) (complete list on Appendix A). Concentrations were measured for individual congeners and aggregated into TEQ values using WHO 2005 toxic equivalency factors (TEFs) [15,16,17]. TEQ values were calculated under three quantification assumptions—assuming 0, 50, or 100% of the lower-bound quantification limit (LQ)—to reflect analytical uncertainty and detection variability [17,50].

### 2.12. Statistical Analysis

Statistical analyses and graphical representations were performed using R (version 4.3.0) in RStudio (version 2025.05.1). Assumptions of normality and homogeneity of variance were evaluated using the Shapiro–Wilk test and Levene’s test, respectively. One-way ANOVA was used to assess differences between treatments when at least three replicates were available. When significant effects were detected (*p* < 0.05), Tukey’s Honestly Significant Difference (HSD) post hoc test was applied for pairwise comparisons.

## 3. Results and Discussion

### 3.1. Enriched Consortia Can Degrade Both DBF and 2,7-DD

The degradation efficiency of the enriched microbial consortia was evaluated at the end of the enrichment phases by measuring the reduction in DBF and 2,7-DD concentrations. Firstly, DBF degradation was assessed after 3 weeks of incubation with 500 mg/L of substrate at the end of the sixth enrichment cycle. As shown in Figure 2A, all bacterial consortia significantly reduced DBF concentrations compared to the abiotic control (CTRL–). The ERA consortium exhibited the highest degradation (32%, *p* < 0.01), followed by AST (22%, *p* < 0.05) and SIG (19%, *p* < 0.05). Although ERA showed the highest mean reduction, no statistically significant differences were observed between the three sites. Given the low number of independent replicates, the observed variation suggests a possible trend, but it is insufficient to support a statistically robust distinction in degradation performance among the consortia.

In a separate test, degradation of 2,7-DD was evaluated by incubating bacterial strains in fresh medium supplemented with 5 mg/L of the compound for 3 weeks. As shown in Figure 2B, all treatments led to measurable degradation relative to the abiotic control, although at lower levels than for DBF. AST showed the highest reduction (12%, *p* < 0.05), followed by SIG (8%, *p* < 0.05), while ERA achieved only a minor reduction (1%, not significant).

These results confirm that consortia enriched from all three sites are capable of degrading both DBF and 2,7-DD, with varying degrees of efficiency. The site-dependent differences observed highlight the relevance of source-specific microbial communities and suggest the potential for tailoring bioremediation strategies to the physicochemical and biological context of the contaminated environment [21,22,29,41].

### 3.2. Five Bacterial Strains Identified for Putatative Degradation Activity

Individual colonies were isolated from TSA plates after the final DBF enrichment step, as described in Section 2.6, and some representative colonies were selected for taxonomical identification using MALDI-TOF mass spectrometry. The complete list of identified bacterial species, along with their respective site of origin and Risk Group classification, is presented in Table 1. Classification was based on Swiss, European, and US biosafety standards. Only strains assigned to biosafety Risk Group 1—defined as non-pathogenic, non-genetically modified organisms posing negligible risk to humans and the environment—were retained for further testing [44,45]. Species with ambiguous taxonomic identification or classified as Risk Group 2 were excluded as a precautionary measure.

The total number of cultivable isolates obtained across all sites was limited, a result consistent with previous studies employing selective enrichment in mineral media [51,52]. These conditions favor the proliferation of a narrow subset of microorganisms capable of metabolizing the selective substrate—in this case, DBF—as the sole carbon and energy source, often resulting in low-diversity but functionally specialized communities [52]. As a result, enrichment often yields low-diversity communities composed of functionally specialized bacteria [53,54]. The dominance of genera such as *Pseudomonas* and *Acinetobacter* is widely reported in the literature related to the degradation of aromatic and halogenated compounds, reflecting their versatile catabolic capabilities and ecological adaptability [55,56]. In this study, however, no clear correlation was observed between the number of cultivable species isolated from each site and the measured degradation efficiency of DBF and 2,7-DD (Figure 2), suggesting that functional performance is driven more by the metabolic traits of dominant strains than by species richness [57,58]. This underscores the relevance of functional specialization over diversity in selective enrichment systems [58].

For example, AST site yielded the highest number of cultivable isolates, but it did not exhibit the highest DBF degradation. In contrast, the ERA consortium showed the most efficient DBF degradation despite the isolation of only two strains. Another limitation lies in the restricted number of strains that can grow on generic culture media, such as those used in this study [59]. While such media may underrepresent total microbial diversity, they are essential for identifying robust and scalable strains suitable for bioremediation. In this context, the use of non-selective but cost-effective media aligns with the operational needs of applied microbial treatments [60,61].

Based on biosafety classification and metabolic relevance, five bacterial strains were selected for degradation assays: *Pseudomonas kermanshahensis* (ERA), *Acinetobacter bohemicus*, *Bacillus velezensis* and *Pseudomonas chlororaphis* (AST), and *Pseudomonas protegens* (SIG). These strains were tested both individually and in defined consortia to assess their ability to degrade DBF and 2,7-DD. Results from these assays are described in the following section.

### 3.3. DBF and 2,7-DD Degradation Efficiency Is Higher in Five-Species Co-Culture Compared to Monocultures

Bacterial monocultures and a defined co-culture were tested for their degradation capabilities toward DBF and 2,7-DD (Figure 3). Degradation analyses were performed after 3 weeks of incubation in mineral medium supplemented with 10 mg/L of both DBF and 2,7-DD. Among the five monocultures, only *A. bohemicus*, *B. velezensis* and *P. chlororaphis*—all isolated from the AST site —demonstrated measurable DBF degradation activity, reducing DBF by 23, 20 and 18%, respectively, compared to the abiotic control (Figure 3A; *p* < 0.05).

Degradation of 2,7-DD was lower, consistent with established knowledge regarding the increased recalcitrance of chlorinated aromatics compared to non-halogenated analogues. Similarly to DBF results, only *A. bohemicus* and *B. velezensis* exhibited significant degradation (8 and 7% reduction, respectively; *p* < 0.05) of 2,7-DD, whereas activity by *P. chlororaphis* remained minimal and not statistically significant (4%, Figure 3B). *P. kermanshahensis* and *P. protegens* showed no measurable degradation activity individually under these conditions, hence they are not shown in Figure 3). None of the three active strains (*A. bohemicus*, *B. velezensis*, *P. chlororaphis*) have been previously reported in the literature for their ability to degrade dioxins or dibenzofurans. However, their respective genera have been widely documented as involved in dioxin degradation through both molecular and functional studies [62,63,64,65,66]. This study thus provides one of the first experimental demonstrations of dioxin-model compound degradation by these species, particularly highlighting *A. bohemicus* and *B. velezensis* as promising candidates for future bioremediation applications.

Interestingly, the five strain co-culture significantly increased degradation for both DBF (50%) and 2,7-DD (27%; Figure 3A,B, *p* < 0.01 in comparison to abiotic control and *p* < 0.05 in comparison to induvial strains). The observed degradation enhancement strongly suggests a synergistic interaction among strains, including those inactive individually. This process could be promoted by complementary metabolic pathways or co-metabolic mechanisms, which have been previously reported in microbial consortia degrading aromatic pollutants [67]. Specifically, certain strains may perform initial transformations, converting complex compounds into intermediates more easily metabolized by others [68,69]. Additionally, strains lacking direct degradation capacity could still contribute indirectly through the secretion of biosurfactants, enzymes, or growth-promoting factors that increase the overall bioavailability and metabolic turnover of the contaminants [67].

Given the substantial effect observed in co-culture, all five strains were retained for subsequent phases of the study, including those that did not exhibit measurable degradation individually. This approach ensures that potential beneficial interactions are preserved and leveraged in future stages of development [67].

### 3.4. Antibiotic Susceptibility Results Do Not Compromise Future Use of the Strains

Antibiotic susceptibility of the five selected bacteria was quantitatively assessed by determining the MIC, expressed as the lowest antibiotic concentration capable of inhibiting visible bacterial growth after incubation. MIC values were measured for 14 antibiotics from different classes (Appendix A).

The results showed variable susceptibility profiles among the isolates. *A. bohemicus* and *B. velezensis* were sensitive to all tested antibiotics, exhibiting MIC values well within the effective range, suggesting a favorable biosafety profile for these two strains and confirming literature findings [70,71]. In contrast, the three *Pseudomonas* isolates exhibited resistance to multiple antibiotics: *P. protegens* was resistant to 7 of the 14 antibiotics tested, including clavulanic acid, ampicillin, clindamycin, erythromycin, linezolid, meropenem, and vancomycin. *P. chlororaphis* was resistant to 5 antibiotics (clavulanic acid, ampicillin, clindamycin, linezolid, vancomycin), while *P. kermanshahensis* showed resistance to clavulanic acid, ampicillin, erythromycin, and linezolid. Most observed resistances were toward antibiotics typically ineffective against Gram-negative bacteria, particularly common among *Pseudomonas* species. Notably, these findings align with existing literature reporting natural intrinsic resistance of environmental *Pseudomonas* strains to several clinically relevant antibiotics (e.g., to macrolides, beta-lactams, glycopeptides, and oxazolidinones) independent of anthropogenic selection pressure [72]. Thus, the resistance profiles detected here likely reflect natural microbial ecology rather than recent anthropogenic influence [73]. Given that all strains are classified as Risk Group 1, posing negligible risk to human health and the environment, and are indigenous to the studied Lausanne soils, the observed antibiotic resistance does not necessarily preclude their use for bioremediation purposes [72]. Nevertheless, these findings underscore the importance of carefully evaluating antibiotic resistance profiles in environmental isolates before considering large-scale bioremediation applications, particularly to address potential biosafety concerns for field operators and broader public health implications [72,74].

### 3.5. The Five Selected Strains Are Suitable for Industrial Scale-Up

Scale-up of selected bacterial strains was carried out in 2 L TSB cultures to assess their suitability for future environmental applications. Scaling to larger volumes is essential for practical bioremediation deployment. However, this process cannot utilize DBF or 2,7-DD as carbon sources due to safety and environmental contamination concerns. Thus, cultivation in generic media such as TSB was required. This shift presents the potential risk of phenotype loss, as bacteria could lose their capacity to degrade specific pollutants when cultured without selective pressure from these substrates [75]. Scale-up culture experiments were performed simultaneously for each strain in biological duplicate for 50 h.

Under these generic conditions, all strains reached high cell densities (10^8^–10^9^ cells/mL) within 24 h, demonstrating efficient growth in TSB (Appendix A). Doubling times varied moderately, with *B.s velezensis* and *A. bohemicus* exhibiting notably short doubling times (1.21 h and 1.27 h, respectively), consistent with efficient biomass production and literature reports (Table 2) [76,77,78]. In contrast, OD600 readings did not always correlate with viable cell counts for certain strains—particularly *P. chlororaphis* and *B. velezensis*—likely due to the production of extracellular pigments or light-absorbing metabolites (Appendix A) [79,80,81].

Cultures maintained stable metabolic activity throughout growth,, with minimal pH fluctuations within a narrow range (6.5–7.4), indicating efficient nutrient utilization without substantial acidification or alkalinization (Appendix A) [82]. Conductivity increased slightly but consistently, reflecting normal metabolic processes and ionic metabolite release (Appendix A) [83]. Quality-control analyses for agri-food domain confirmed the absence of pathogenic bacterial contaminants, ensuring biosafety for potential environmental use (Appendix A).

Based on these results, it can be concluded that scale-up of these strains is feasible under controlled conditions. Bacterial concentration (expressed as cells/mL) and conductivity emerged as the most reliable indicators of bacterial proliferation among the tested parameters.

### 3.6. Phenotype Stability Is Preserved After Large Biomass Production

Phenotype stability is the retention of pollutant-degrading capability regardless of presence/absence of target pollutant and it is a crucial parameter to test for a successful scale-up. Phenotype stability was evaluated following biomass production in non-selective TSB medium. To test this, bacterial strains previously scaled-up in TSB were re-inoculated into mineral medium supplemented with 2,7-DD (10 mg/L) as sole carbon sources and incubated for 4 weeks. Degradation results confirmed the preservation of the pollutant-degrading phenotype for most isolates, albeit with varying efficiencies among strains and compounds (Figure 4).

Specifically, *A. bohemicus* retained significant degradation activity for 2,7-DD, showing 38% reduction, thus confirming its robustness and strong catabolic activity previously observed in individual strain tests (Figure 3B). Conversely, *P. chlororaphis* exhibited a complete loss of measurable 2,7-DD degradation capability, indicating phenotype instability after cultivation in TSB without selective pressure. This result contrasts with earlier observations (Figure 3B), where this strain showed modest but detectable degradation activity.

*P. protegens* and *P. kermanshahensis* demonstrated measurable 2,7-DD reduction (17% and 9%, respectively) despite previously exhibiting limited or negligible degradation capacity. This result suggests that nutrients provided by the TSB medium—such as amino acids and other organic compounds—may have contributed to activating additional metabolic pathways or enhancing enzymatic activity involved in pollutant degradation, as suggested by previous studies [84,85,86]. Lastly, *B. velezensis* showed substantial degradation activity (23%), demonstrating considerable catabolic potential for chlorinated aromatics event after growth under unselective conditions.

The co-culture containing all selected strains—including those not individually active—again showed enhanced degradation capacity (20%) compared to most monocultures. However, this increase was not statistically significant when compared to the individual strains, in contrast to the results presented in Figure 3B. One possible explanation is that growth in TSB medium may have strengthened the metabolic capabilities of individual strains [84,85,86], thereby diminishing the relative impact of potential synergistic interactions when the strains were subsequently tested in co-culture [58]. Nevertheless, the co-culture still exhibited consistent pollutant degradation, confirming the functional stability and overall utility of defined microbial consortia for aromatic and chlorinated pollutant bioremediation applications.

Degradation efficiency normalization per bacterial concentration after 3 weeks (Table 3) highlighted *B. velezensis* clearly as the most effective strain, significantly outperforming all others (57.9% per 1 × 10^7^ cells/mL), followed by *A. bohemicus and P. protegens* (approximately 1.0% per 1 × 10^7^ cells/mL each). The enhanced degradation capability of *B.s velezensis* in terms of biomass-normalized efficiency suggests that this strain may be particularly effective and advantageous for practical bioremediation scenarios.

In summary, these findings confirm that most bacterial strains retain their degrading phenotype after cultivation in non-selective medium, with substantial variability among strains. Importantly, the pronounced degradation activity in co-culture and by *B. velezensis* highlights the significance of maintaining diverse bacterial consortia for bioremediation applications, thereby providing functional redundancy and improved biodegradation performance under environmental conditions [58].

### 3.7. Biodegradation Performance of Selected Bacterial Treatments in Real Soil Samples

To evaluate the degradation potential of selected strains under semi-natural conditions, two treatments were implemented based on laboratory-scale performance. Samples were monitored over a six-month period with time points at 4, 8 and 24 weeks. Treatment 1 consisted of the two most effective isolates (*A. bohemicus* and *B. velezensis*), previously identified for their consistent degradation of model compounds. Treatment 2 employed consortium of all selected strains: *A. bohemicus*, *B. velezensis*, *P. protegens*, *P. kermanshahensis*, and *P. chlororaphis*, aiming to leverage complementary metabolic activities. Repeated measurements of dioxin concentrations in untreated soil samples from the ERA, SIG, and AST sites revealed considerable variability across congeners, with relative changes ranging from approximately 5% to 37%, depending on the compound and site (Appendix A). This variability likely reflects a combination of factors, including intrinsic heterogeneity in soil composition, uneven distribution of contaminants at the microscale, and the known complexity of organic pollutant–matrix interactions [87,88]. Analytical uncertainty—especially relevant at lower concentrations—further contributes to dispersion in measured values [89,90]. These elements must be carefully considered when assessing whether observed changes are attributable to microbial activity or simply fall within expected variability [91]. Absolute concentrations of each congener in each site are reported in Appendix A, while Appendix A summarizes relative variation (%) for each treatment, site, and time point compared to controls. Notably, untreated controls did not exhibit any appreciable decrease in dioxin levels over time, confirming the absence of natural attenuation within the study window.

Overall, no statistically significant differences were detected between treatments and controls (*p* > 0.05), reflecting the variability already evident in untreated soils (Appendix A). Some tendencies emerged: early reductions were mainly associated with Treatment 1 at ERA and AST, while at 8 weeks this consortium appeared more consistent at SIG. By 24 weeks, reductions diminished overall compared to the reductions measured after 8 weeks, with measurable effects only in SIG. This decline may reflect microbial attenuation, reduced pollutant bioavailability, or stronger sorption of dioxins to soil organic matter [21,88,90,92]. Overall, the results indicate that treatment effects were transient, site-dependent, and not cumulative, with the two-strain consortium generally performing more consistently than the broader co-culture.

## 4. Conclusions

This study demonstrates that indigenous bacterial strains isolated from chronically contaminated urban soils can degrade dioxin-like compounds, even when obtained through relatively simple microbiological techniques. Defined co-cultures enhanced degradation efficiency under highly selective conditions—such as mineral media with dioxins or DBF as sole carbon sources—likely due to metabolic complementarity and mutual support among strains. However, this advantage diminished under nutrient-rich or real-soil conditions, suggesting that the role of co-cultures may be primarily nutritional support under extreme resource limitation [84,85,86].

Biomass production could be successfully scaled up in non-selective media while preserving the degradation phenotype, which represents an important operational step for future applications. At the same time, the study highlighted a discrepancy between laboratory-based assays and soil microcosm experiments, likely due to multiple factors, including pollutant bioavailability [19,21], soil matrix interactions [92], ecological competition [93], and analytical uncertainty [23,92].

Importantly, the use of field-oriented microcosm setups proved more informative than artificial conditions, supporting the consensus that in situ or near-field conditions are essential for a realistic evaluation of bioremediation strategies [38,40,94]. Notably, *A. bohemicus* and *B. velezensis* are reported here for the first time as active in dioxin and dibenzofuran degradation, making them promising candidates for future bioremediation applications. Looking ahead, applying these bacterial treatments to full-scale test plots would enable robust statistical assessment and repeated treatment cycles, helping to establish realistic degradation kinetics and evaluate feasibility in urban or redevelopment contexts, where time efficiency is often a major limitation of bioremediation technologies [21,38,95].

## Figures and Tables

**Figure 1 microorganisms-13-02306-f001:**
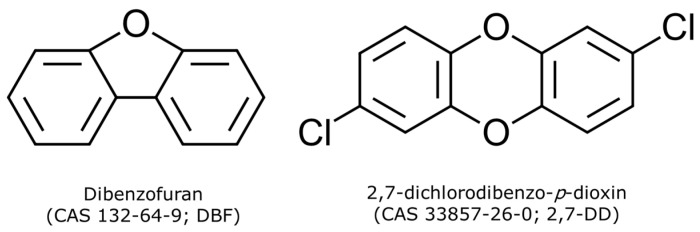
Chemical structures of the two model compounds used in this study: dibenzofuran (DBF, **left**) and 2,7-dichlorodibenzo-*p*-dioxin (2,7-DD, **right**).

**Figure 2 microorganisms-13-02306-f002:**
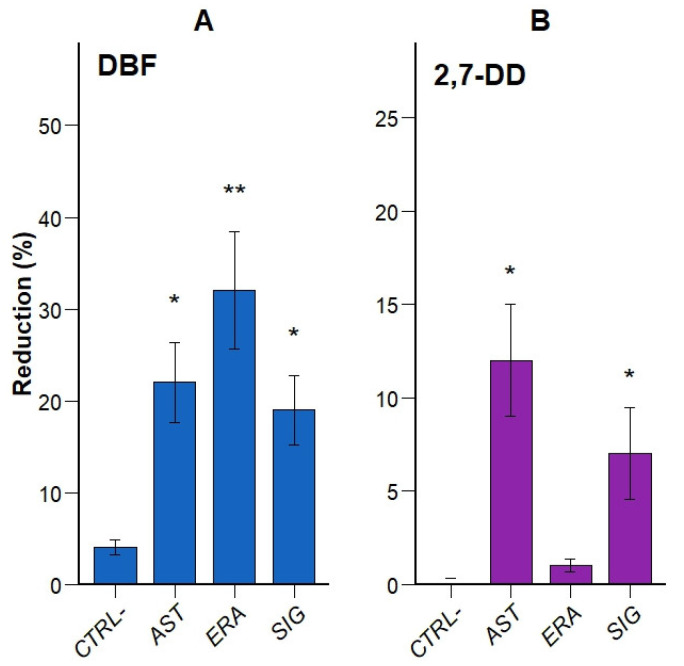
Percentage reduction in dibenzofuran (DBF, panel (**A**)) and 2,7-dichlorodibenzo-*p*-dioxin (2,7-DD, panel (**B**)) after 3 weeks of incubation with enriched microbial cultures derived from three soil sampling sites: Ancien Stand (AST), Eracom (ERA), and Signal (SIG). Initial concentrations were 500 mg/L for DBF and 5 mg/L for 2,7-DD, separately. Results are compared to an abiotic negative control (CTRL–). * *p* < 0.05, ** *p* < 0.01, *n* = 3.

**Figure 3 microorganisms-13-02306-f003:**
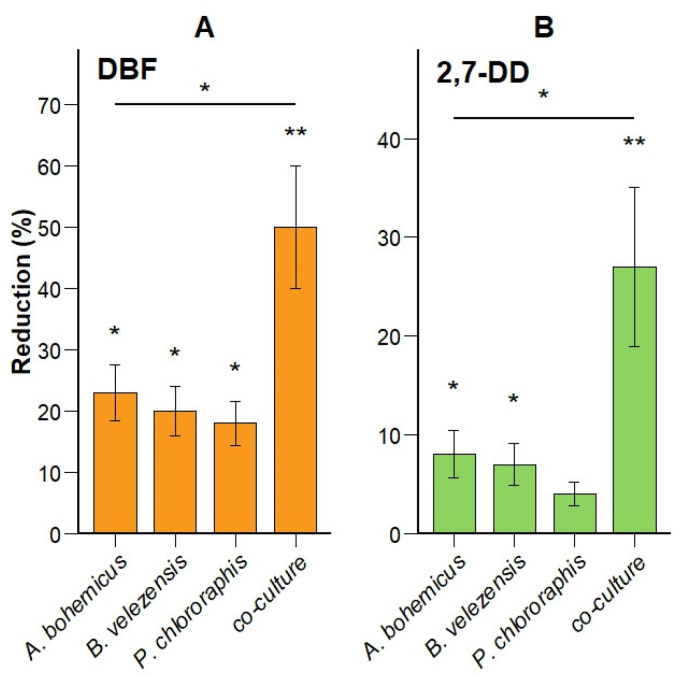
Percentage reduction in dibenzofuran (DBF, panel (**A**)) and 2,7-dichlorodibenzo-*p*-dioxin (2,7-DD, panel (**B**)) after 3 weeks of incubation (initial concentration 10 mg/L) with individual strains or a defined synthetic bacterial co-culture. Only strains demonstrating measurable activity individually (*A. bohemicus*, *P.s chlororaphis*) are shown. Co-culture: five-strain co-culture.* *p* < 0.05, ** *p* < 0.01.

**Figure 4 microorganisms-13-02306-f004:**
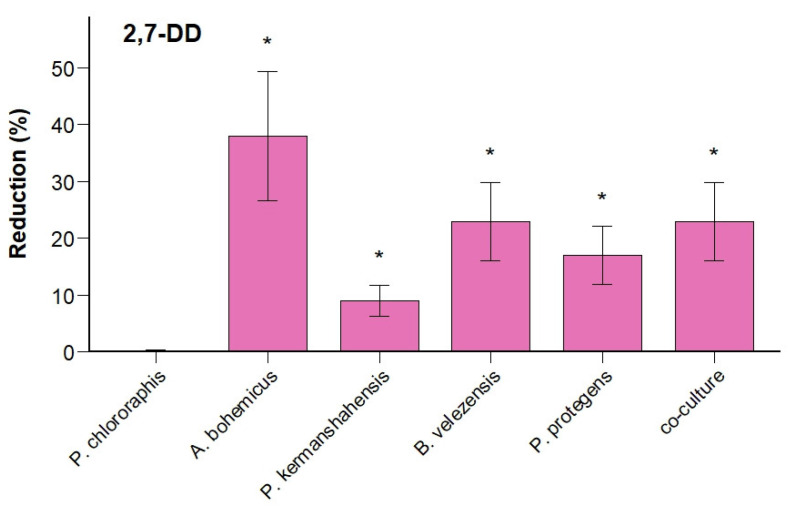
Degradation of 2,7-DD by individual strains and defined co-culture following biomass production in TSB. Strains were incubated for 4 weeks in mineral medium containing 10 mg/L of 2,7-DD as the sole carbon source. Co-culture: five-strain co-culture.* *p* < 0.05.

**Table 1 microorganisms-13-02306-t001:** Bacterial strains isolated from enrichment cultures and their biosafety classification. Strains were identified via MALDI-TOF and classified according to national and international microbial risk guidelines.

Site	Identified Species	Risk Group
ERA	*Pseudomonas kermanshahensis*	1
ERA	*Stenotrophomonas nematodicola*	ND
AST	*Acinetobacter bohemicus*	1
AST	*Bacillus velezensis*	1
AST	*Pseudomonas shirazica*	ND
AST	*Achromobacter aegrifaciens*	2
AST	*Pseudomonas chlororaphis*	1
AST	*Stenotrophomonas genomosp.*	2
SIG	*Pseudomonas protegens*	1

ND: not classified. ERA: Eracom, AST: Ancien Stand, SIG: Signal.

**Table 2 microorganisms-13-02306-t002:** Doubling times of selected bacterial strains during scale-up in 2 L TSB cultures. Values represent mean doubling times calculated from duplicate cultures per strain.

Specie	Doubling Time (h)
*Acinetobacter bohemicus*	1.27
*Bacillus velezensis*	1.21
*Pseudomonas chlororaphis*	3.32
*Pseudomonas kermanshahensis*	2.03
*Pseudomonas protegens*	1.50

**Table 3 microorganisms-13-02306-t003:** Degradation efficiency of 2,7-DD normalized per 1 × 10^7^ cells/mL in mineral medium with 10 mg/L 2,7-DD. Strains were transferred in mineral medium with 10 mg/L 2,7-DD after biomass production in TSB. Values include an estimated ±20% error margin.

Specie/Condition	% Degradation per 1 × 10^7^ Cells/mL
*Pseudomonas chlororaphis*	ND
*Acinetobacter bohemicus*	1.0%
*Pseudomonas kermanshahensis*	0.5%
*Bacillus velezensis*	57.9%
*Pseudomonas protegens*	1.0%
Co-culture (all strains)	1.7%

ND: no degradation.

## Data Availability

The data supporting the findings of this study are available from the corresponding author upon request.

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
