# Peer review of "Degradation of Dioxins and DBF in Urban Soil Microcosms from Lausanne (Switzerland): Functional Performance of Indigenous Bacterial Strains"

_microorganisms, 2025, doi:10.3390/microorganisms13102306_

Round 1

Reviewer 1 Report

Comments and Suggestions for Authors

The authors present an interesting paper that is relevant both from the point of view of science and environmental protection. A fairly detailed Introduction introduces readers to the problem. Not only the effects in laboratory conditions were studied, but also the production and preservation of the properties of strains in large flasks, and micro-natural tests were carried out in vessels with soil. It is also important to select exclusively non-pathogenic bacteria and study their resistance to antibiotics.
When reading the manuscript, I had a number of questions:
Line 204-206: "After collecting 10 mL of culture as previously described, the entirety of the residual sample (190 mL) was extracted three times with 10 mL of ethyl acetate each time". Was the liquid sample extracted?
Sections 2.7, 2.11. What volume of culture was added? There is no information on this.
Line 310: "Bioremediation treatments were conducted using soil samples collected over the second sampling campaign from three contaminated sites (SIG, AST, ERA)". Did these soils contain indigenous strains or were they sterilized?
Lines 453-455: "The observed degradation enhancement strongly suggests a synergistic interaction among strains, including those inactive individually". What was the volume of each individual strain added and what was the volume of all strains combined?
Line 473 - extra dash at the end of the sentence.

Author Response

The authors present an interesting paper that is relevant both from the point of view of science and environmental protection. A fairly detailed Introduction introduces readers to the problem. Not only the effects in laboratory conditions were studied, but also the production and preservation of the properties of strains in large flasks, and micro-natural tests were carried out in vessels with soil. It is also important to select exclusively non-pathogenic bacteria and study their resistance to antibiotics.

Answer:

We sincerely thank the reviewer for the positive and encouraging feedback. We are pleased that the scientific and environmental relevance of our work, as well as the methodological choices were appreciated. Such recognition is highly motivating for us and reinforces our commitment to further developing this line of research.

When reading the manuscript, I had a number of questions:

Line 204-206: "After collecting 10 mL of culture as previously described, the entirety of the residual sample (190 mL) was extracted three times with 10 mL of ethyl acetate each time". Was the liquid sample extracted?

Answer:

We thank the reviewer for this observation and apologize if the description was not sufficiently clear. The extracted liquid corresponds to the culture medium. After several laboratory trials, it became evident that extracting the entirety of the culture medium was necessary to obtain reproducible and reliable results. For this reason, 10 mL of culture were collected for microbiological analyses, while the remaining volume (200 mL total minus 10 mL) was used for chemical analyses. The revised text reads as follows:

"From the initial culture volume of 200 mL, 10 mL were collected for microbiological analyses, while the remaining 190 mL were extracted three times with 10 mL of ethyl acetate each time."

Sections 2.7, 2.11. What volume of culture was added? There is no information on this.

Answer:

We sincerely thank the reviewer for this valuable comment and for drawing our attention to the lack of clarity regarding the culture volumes used in Sections 2.7 and 2.11. We apologize for this omission. The missing details have now been added to the revised manuscript as follows:

Section 2.7 (Degradation Assays with Individual Strains and Defined Co-Cultures):

The total assay volume was 20 mL. For monoculture experiments, each strain was inoculated with 200 μL of cell suspension at 10⁹ CFU/mL, while for co-culture experiments 40 μL of each strain suspension at 10⁹ CFU/mL were added (total 200 μL), maintaining the same overall inoculum density of 10⁷ cells/mL.

The revised text now reads:

2.7 Degradation Assays with Individual Strains and Defined Co-Cultures

Biodegradation tests were conducted to assess DBF and degradation capacity of the selected strains in both mono- and co-culture. All assays were performed in sterile mineral medium supplemented with DBF and 2,7-DDat concentration of 5 mg/L each. Cultures were incubated for 3 weeks at 20°C under agitation (130 rpm). The total assay volume was 20 mL. For monoculture experiments, each strain was inoculated with 200 μL of cell suspension at 10⁹ CFU/mL, while for co-culture experiments 40 μL of each strain suspension at 10⁹ CFU/mL were added (total 200 μL), maintaining the same overall inoculum density of 10⁷ cells/mL. DBF and 2,7-DD degradation were quantified by GC-MS at the end of the incubation period.

Section 2.11 (Treatment of Real Soil Samples for Bioremediation Experiments):

Depending on the treatment, inoculation was performed with 250 mL of cell suspension at 10⁸ CFU/mL for the two-strain culture or 100 mL of cell suspension at 10⁸ CFU/mL for the five-strain co-culture. Both conditions reached a final concentration of 10⁷ CFU/g soil.

The revised text now reads:

“2.11 Treatment of Real Soil Samples for Bioremediation Experiments

Bioremediation treatments were conducted using non-sterilized soil samples col-lected over the second sampling campaign from three contaminated sites (SIG, AST, ERA). For each experimental condition, three independent replicates were employed, each consisting of 10 kg of soil per site. Additionally, one untreated 10 kg sample per each site served as negative control. Prior to treatment application, each 10 kg soil sample was thoroughly homogenized using an electric mixer suitable for this scale. Two experimental treatments were conducted: the first involved inoculation with 250 mL of cell suspension at 10⁸ CFU/mL of the two bacterial strains exhibiting the highest degradation performance, while the second treatment consisted of 100 mL of cell sus-pension at 10⁸ CFU/mL of a defined five-strain co-culture. Both treatments were ap-plied at a final concentration of 10⁷ CFU/g soil. (...)”

Line 310: "Bioremediation treatments were conducted using soil samples collected over the second sampling campaign from three contaminated sites (SIG, AST, ERA)". Did these soils contain indigenous strains or were they sterilized?

Answer:

We thank the reviewer for this important point. The soils used in the bioremediation treatments were not sterilized, in order to (i) preserve the indigenous microbial communities and allow the evaluation of potential interactions between native strains and the inoculated consortium, and (ii) avoid chemical alterations or degradation of soil quality that would inevitably result from sterilization. This choice was made to maintain realistic environmental conditions and to ensure the ecological validity of the study.

The text has been modified accordingly and now reads as follows:

"Bioremediation treatments were conducted using non-sterilized soil samples collected over the second sampling campaign from three contaminated sites (SIG, AST, ERA).

Lines 453-455: "The observed degradation enhancement strongly suggests a synergistic interaction among strains, including those inactive individually". What was the volume of each individual strain added and what was the volume of all strains combined?

Answer:

We thank the reviewer for this useful question and apologize for the missing information in the original version. The volumes used in this experiment are the same as those now specified in Section 2.7. Specifically, the total degradation test volume was 20 mL. For monoculture experiments, each strain was inoculated with 200 μL of cell suspension at 10⁹ CFU/mL. For co-culture experiments, 40 μL of each strain suspension at 10⁹ CFU/mL were added (total 200 μL), ensuring the same overall inoculum density of 10⁷ cells/mL.

This clarification has been added to the revised manuscript to improve methodological transparency.

Line 473 - extra dash at the end of the sentence.

Answer:

We thank the reviewer for pointing out this typographical issue. The extra dash at the end of the sentence in Line 473 has been removed in the revised manuscript.

Reviewer 2 Report

Comments and Suggestions for Authors

The manuscript describes the isolation of five bacterial strains capable of degrading model pollutant molecules dibenzofuran (DBF) and 2,7-dichlorodibenzo-p-dioxin (2,7-DD). These bacteria can degrade this molecules with degradation ranges around 15-25% for DBF (10 mg/L) and around 5-10% for 2,7-DD (5 mg/L) in mineral medium, while the use of the consortium composed by all five bacterial isolates, enhances pollutant degradation to 50% for DBF and 27% for 2,7-DD. However, the result of the microcosm soil treatment did not show differences respect to control. All bacteria showed minimal antibiotic resistance that in conjunction with their degradation profiles highlighted potential for possible dioxin-like pollution in soils.

The authors need to address the following commentaries:

The submitted version of the manuscript seems preliminary, due to include commentaries, maybe of the co-authors or reviewers.

Line 23, complement “Risk Group 1” with information of the regulation that stablish the risk groups

Line 60, eliminate extra space in “Convention  [11,12].”

Line 62, eliminate extra space in “[13].  Soil”

Line 69, eliminate period before references [15-17]

Line 116, could be better use “was” instead “is”

Line 118, complement “Risk Group 1” with information of the regulation that stablish the risk groups

Line 198, eliminate extra space in “2,7-DD.  After 3 weeks”

Line 213 and 214, “ppm” could be better us mg/L

Line 252, correct “2,7-DDat  concentration”

Line 273, correct “strain., Bacterial sensitivity”

Line 327, could be better “4, 8, and 24 weeks”

Line 337, could be better “ 0, 50, or 100%”

Line 345, complement with information of the test used to evaluate the normality and homogeneity of the data, describe before the information of the variance analysis

Line 433, correct “B.s velezensis

Line 435, could be better “23, 20, an 18%”

Line 441, , could be better “8 and 7%”

Line 442, eliminate extra space in “Figure 3B).  P. ker”

Line 488, correct “: P.s protegens”

Line 559, Correct “P.s kermanshahensis

In figure 4, use italics for scientific names

586, eliminate extra space in “clearly  as the”

Line 656-686, It is important to generate a novel conclusions section, in the current version of the manuscript conclusion seem discussion

Author Response

The manuscript describes the isolation of five bacterial strains capable of degrading model pollutant molecules dibenzofuran (DBF) and 2,7-dichlorodibenzo-p-dioxin (2,7-DD). These bacteria can degrade this molecules with degradation ranges around 15-25% for DBF (10 mg/L) and around 5-10% for 2,7-DD (5 mg/L) in mineral medium, while the use of the consortium composed by all five bacterial isolates, enhances pollutant degradation to 50% for DBF and 27% for 2,7-DD. However, the result of the microcosm soil treatment did not show differences respect to control. All bacteria showed minimal antibiotic resistance that in conjunction with their degradation profiles highlighted potential for possible dioxin-like pollution in soils.

Answer:

We sincerely thank the reviewer for the careful evaluation and the constructive summary of our work. We are pleased that both the isolation of the five bacterial strains and their degradation performance, as well as the consortium effect, were clearly recognized. We also appreciate the reviewer’s acknowledgment of the antibiotic resistance profiles and the relevance of these findings for potential applications in dioxin-like soil pollution.

The authors need to address the following commentaries:

The submitted version of the manuscript seems preliminary, due to include commentaries, maybe of the co-authors or reviewers.

Line 23, complement “Risk Group 1” with information of the regulation that stablish the risk groups

Answer:

We thank the reviewer for this helpful comment. Since the sentence appears in the Abstract, we considered that adding regulatory details there would weigh down the text. Therefore, we revised the sentence as follows:

"Using selective enrichment techniques, five strains were isolated, with no biosafety concerns for human health and environmental applications."

The regulatory information regarding Risk Group classification has been retained and described explicitly in Section 2.6 (Quantification, Isolation and Identification of Degrading Microbial Strains).

Line 60, eliminate extra space in “Convention  [11,12].”

Line 62, eliminate extra space in “[13].  Soil”

Line 69, eliminate period before references [15-17]

Line 116, could be better use “was” instead “is”

Answer:

We sincerely thank the reviewer for the detailed revision. All the comments have been carefully assessed and the suggested corrections have been implemented in the revised manuscript:

Line 60: extra space in “Convention [11,12].” removed.

Line 62: extra space in “[13]. Soil” corrected.

Line 69: period before references [15–17] eliminated.

Line 116: verb corrected to “was” instead of “is”.

Line 118, complement “Risk Group 1” with information of the regulation that stablish the risk groups

Answer:

We thank the reviewer for this comment. The text has been corrected to explicitly indicate the regulatory framework defining the microbial risk groups. The revised sentence now reads:

"It involves the use of bacterial strains strictly belonging to Risk Group 1 according to EU and US classification, defined as organisms posing negligible or no risk to human health or the environment."

Line 198, eliminate extra space in “2,7-DD.  After 3 weeks”

Line 213 and 214, “ppm” could be better us mg/L

Line 252, correct “2,7-DDat  concentration”

Line 273, correct “strain., Bacterial sensitivity”

Line 327, could be better “4, 8, and 24 weeks”

Line 337, could be better “ 0, 50, or 100%”

Answer:

We thank the reviewer for these careful observations. All the indicated points have been corrected in the revised manuscript as follows:

Line 198: extra space in “2,7-DD. After 3 weeks” removed.

Lines 213–214: unit ppm replaced with mg/L.

Line 252: corrected to “2,7-DD at concentration”.

Line 273: corrected to “strain. Bacterial sensitivity”.

Line 327: revised to “4, 8, and 24 weeks”.

Line 337: revised to “0, 50, or 100%”.

Line 345, complement with information of the test used to evaluate the normality and homogeneity of the data, describe before the information of the variance analysis

Answer:

We thank the reviewer for this useful comment. The requested information regarding the tests used to evaluate normality and homogeneity of variance has now been added, and the description has been moved before the information on variance analysis for improved clarity

2.12 Statistical Analysis

Statistical analyses and graphical representations were performed using R (ver-sion 4.3.0) in RStudio (version 2025.05.1). Assumptions of normality and homogeneity of variance were evaluated using the Shapiro–Wilk test and Levene’s test, respectively. One-way ANOVA was used to assess differences between treatments when at least three replicates were available. When significant effects were detected (p < 0.05), Tuk-ey’s Honestly Significant Difference (HSD) post hoc test was applied for pairwise comparisons.

Line 433, correct “B.s velezensis

Line 435, could be better “23, 20, an 18%”

Line 441, , could be better “8 and 7%”

Line 442, eliminate extra space in “Figure 3B).  P. ker”

Line 488, correct “: P.s protegens”

Line 559, Correct “P.s kermanshahensis

Answer:

We thank the reviewer for the careful revision. All the suggested corrections have been implemented in the revised manuscript:

Line 433: corrected “B.s velezensis”.

Line 435: revised to “23, 20, and 18%”.

Line 441: revised to “8 and 7%”.

Line 442: extra space in “Figure 3B). P. ker” removed.

Line 488: corrected “: P.s protegens”.

Line 559: corrected “P.s kermanshahensis”.

In figure 4, use italics for scientific names

Answer:

We thank the reviewer for this observation and we fully agree in principle. However, in this specific case the choice not to italicize the scientific names was made for graphical reasons, due to the software (IMiGraph) and the chosen presentation style. When formatted in italics, the names became less legible in the figure layout. The same consideration applies to Figure 3.

If the editor has no objections and if acceptable to the reviewer, we would prefer to maintain the current formatting for clarity.

586, eliminate extra space in “clearly  as the”

Answer:

We thank the reviewer for this careful observation. The extra space in “clearly as the” has been removed in the revised manuscript.

Line 656-686, It is important to generate a novel conclusions section, in the current version of the manuscript conclusion seem discussion

Answer:

We thank the reviewer for this important comment. We agree that in the original version, the Conclusions section contained excessive discussion and did not fully highlight the main take-home messages. We have therefore rewritten this section in a more concise and conclusive style, focusing on the novel findings and their implications. References have been removed in this section, as recommended by the journal guidelines, and are now only cited in the main text.

The revised Conclusions now read as follows:

This study demonstrates that indigenous bacterial strains isolated from chroni-cally contaminated urban soils can degrade dioxin-like compounds, even when ob-tained through relatively simple microbiological techniques. Defined co-cultures en-hanced degradation efficiency under highly selective conditions—such as mineral me-dia with dioxins or DBF as sole carbon sources—likely due to metabolic complementa-rity and mutual support among strains. However, this advantage diminished under nutrient-rich or real-soil conditions, suggesting that the role of co-cultures may be primarily nutritional support under extreme resource limitation [84-86].

Biomass production could be successfully scaled up in non-selective media while preserving the degradation phenotype, which represents an important operational step for future applications. At the same time, the study highlighted a discrepancy between laboratory-based assays and soil microcosm experiments, likely due to multi-ple factors, including pollutant bioavailability [19,21], soil matrix interactions [92], ecological competition [93], and analytical uncertainty [23,92].

Importantly, the use of field-oriented microcosm setups proved more informative than artificial conditions, supporting the consensus that in situ or near-field conditions are essential for a realistic evaluation of bioremediation strategies [38,40,94]. Notably, A. bohemicus and B. velezensis are reported here for the first time as active in dioxin and dibenzofuran degradation, making them promising candidates for future bioremediation applications. Looking ahead, applying these bacterial treatments to full-scale test plots would enable robust statistical assessment and repeated treatment cycles, helping to establish realistic degradation kinetics and evaluate feasibility in urban or redevelopment contexts, where time efficiency is often a major limitation of bioremediation technologies [21,38,95].

Reviewer 3 Report

Comments and Suggestions for Authors

Please, standardize the units: (ng/kg) /replace by:  (ngkg-1

3 g/L / 3 gL-1  along the manuscript and in figure legends

Authors show indigenous bacterial strains isolated from chronically contaminated urban soils with measurable dioxin-degrading activity, even when obtained using  accessible microbiological techniques.  confirming that defined co-cultures tend to enhance degradation efficiency in highly 
selective conditions—such as mineral media with dioxins or DBF as sole carbon sources

well presented, with 2 tables and 3 figures, thus too short.

Author Response

Authors show indigenous bacterial strains isolated from chronically contaminated urban soils with measurable dioxin-degrading activity, even when obtained using  accessible microbiological techniques.  confirming that defined co-cultures tend to enhance degradation efficiency in highly  selective conditions—such as mineral media with dioxins or DBF as sole carbon sources

Answer:

We sincerely thank the reviewer for the time dedicated to carefully reading our manuscript and for the favourable feedback.

Please, standardize the units: (ng/kg) /replace by:  (ngkg-1 3 g/L / 3 gL-1  along the manuscript and in figure legends

Answer:

We thank the reviewer for drawing our attention to this issue and apologize for the oversight. All units throughout the manuscript and figure legends have now been standardized in the requested format using the slash notation (e.g., ng/kg, 3 g/L).

well presented, with 2 tables and 3 figures, thus too short.

Answer:

We sincerely thank the reviewer for this comment. Our manuscript in its current form presents 4 figures and 3 tables in total, which positions it in the medium-to-upper range of figure/table content typically published in this journal. To avoid making the main text overly long and less readable, we decided to include 6 additional tables and 4 figures in the Supplementary Information. This aspect has now been further emphasized in the revised manuscript following the reviewer’s suggestion. We would also be happy to move selected figures or tables from the Supplementary Information to the main text if this is preferred by the reviewer or the editor, and we will gladly follow their recommendation in this respect.

Round 2

Reviewer 2 Report

Comments and Suggestions for Authors

After reviewing the manuscript for a second time, I believe the authors have fully complied with all of the reviewers' recommendations.

Therefore, I believe that in its current state, it can be accepted for publication.